# The impact of digital technologies on knowledge networks in two engineering organisations

**Eann A. Patterson[1], Richard J. Taylor[2], Yuxin Yao[3]***

**1** School of Engineering, University of Liverpool, Liverpool, United Kingdom, **2** Department of Mechanical, Aerospace and Civil Engineering, University of Manchester, Manchester, United Kingdom, **3** School of Engineering, University of Manchester, Manchester, United Kingdom

* yaoyx9624@gmail.com

**Data Availability Statement:** All relevant data are within the paper and its Supporting Information files.

**Funding:** This research was jointly funded by the EPSRC Centre for Doctoral Training in NGN (Next

## Abstract

The management and exploitation of knowledge can contribute to the competitive advantage of an organisation and hence can be a driver of its value. This paper examines knowledge management practices within two engineering organisations with an emphasis on barriers to its effectiveness and the influence of current and anticipated deployment of digital technologies. Two methods were used to gather research data across both organisations by combining a series of semi-structured interviews with a quantitative social network analysis. Examination of the acquired data provided insights into the relationship between the knowledge management culture in both organisations and their adoption of digital tools. Hudson's evolutionary model of safety culture has been modified to describe the culture of knowledge management in an organisation and the acquired data used to locate the two organisations on their knowledge management journey. It is proposed that social networks could be used as an indicator of the stage of evolution of knowledge management in engineering organisations more generally with low network densities and dispersed networks representing higher stages of evolution.

## 1. Introduction

When an organisation is a social community specialising in the speed and efficiency of the creation and transfer of knowledge [1] then knowledge is the foundation of an organisation's competitive advantage and the primary driver of an organisation's value [2]. More details on concepts of data, information and knowledge, types of knowledge, and knowledge processes can be found in, for example, [3–5]. Many knowledge intensive organisations have invested in tools and policies to foster knowledge sharing and to solve knowledge sharing problems [6]. It is widely recognised that poor communication and coordination within an established network leads to poor organisational performance [7] and therefore, effective knowledge management is of vital importance within engineering organisations whose functions involve the planning, design, processing and delivery of engineering products or services. Digital technologies are providing an expanding box of tools to support knowledge management; however,

Generation Nuclear) and the University of Manchester. The funders had no role in study design, data collection and analysis, decision to publish, or preparation of the manuscript.

**Competing interests:** The authors have declared that no competing interests exist.

there is evidence in the literature that the implementation of these tools is not uniform and rarely holistic [8]. Hence, it is relevant to examine the extent to which the adoption of digital technologies is supporting the knowledge management culture of knowledge intensive organisations.

A network approach is used in knowledge management as they both deal with the concept of interactivity [9]. Interactions among people are generated through communication, coordination and collaboration. Nonaka and Takeuchi (1995) likened knowledge to a fluid where these interactions can be represented as flows of information and knowledge [10]. The way that knowledge flows in organisations is often a hidden process [11]. A social network analysis can be used to map and analyse these invisible knowledge flows.

Social network analysis (SNA) is a sociological paradigm used to analyse structural patterns in relationships amongst individuals, groups and organisations [12]. It allows researchers to examine naturally existing patterns and relationships between and within groups, which is especially useful for studying human behaviour [13]. Some researchers have employed social network analysis to understand knowledge sharing and knowledge creation within organisations [14–16]. However, despite promising developments in some fields, SNA is still underused in the engineering field, and there is nearly no published research in which SNA has been applied to analyse engineering knowledge management [8].

This paper examines knowledge management practices within two engineering organisations by addressing the following research questions: 1. How is knowledge being communicated within and across engineering organisations? 2. What are the current challenges to knowledge management? 3. What are the expected improvements in knowledge management as a result of the use of digital technology? Through a two-method approach combining social network analysis and semi-structured interviews, two case studies were conducted in different engineering organisations. Based on the acquired data, this paper assesses the implications of the adoption of digital technologies on current and future engineering knowledge management practices and introduces a model for the evolution of a knowledge management culture based on an established model for health and safety culture in organisations [17].

## 2. Research methodology

### 2.1 Case study design

Two engineering organisations with differing digital maturity and organisational cultures were chosen for the case studies. Organisation A is a national and international engineering consultancy company. A three-month ethnographic study was conducted in one of their satellite offices, which was delivering infrastructure projects for decommissioning and construction clients in the area. In total, 31 interviews were conducted in Organisation A.

Organisation B is a company that generates electricity. A series of semi-structured interviews were conducted within a one-month period. In total, 16 interviews were conducted in Organisation B. There were no known interactions between the two organisations.

### 2.2 Mixed method social network analysis

Social network analysis can be used to understand the structural and relational characteristics of knowledge networks [18]. Mixed-method social network analysis involves combining SNA methods with other more qualitative approaches [19]. A mixed-method approach enables researchers to both map and measure network properties and to explore and explain issues relating to the generation, variability and dynamics of network ties, and the meaning involved in those ties.

For use in both organisations, a survey was created based on the social network analysis methodology used by Knoke and Yang (2019) in their study [18]. An ego-centric approach was used to generate the social networks. In ego-centric studies, a set of focal nodes called "egos" and their ties to other nodes called "alters" are assessed [20]. The "alters" are not necessarily among the set of egos. The selection of the interviewees followed the maximum-variation strategy [15]. Interviewees were selected to provide a distribution of official roles, length of experience working in the organisation, and areas of expertise. Before selecting interviewees in each organisation, meetings with senior staff in two organisations were conducted to understand their organisational culture and the role of people in the organisational structure. The set of interviewees were asked to fill in the survey which was sent to them seven days before conducting the interviews. Two questions were asked in the survey about who sought knowledge from the 'egos' and about from whom the 'egos' sought knowledge. The answers to these questions were used to map two types of knowledge transfer network that exist in the organisations, one describing the flow of knowledge towards the 'egos' and the other away from the 'egos'. A third question was asked about with whom the 'egos' interacted when new knowledge was needed and these responses were used to map the knowledge creation network in each organisation describing the knowledge flows associated with generating new knowledge. 25 individuals (81% of those canvassed) in Organisation A returned the surveys, and 12 people (75%) in Organisation B, which was considered an adequate response because a network response rate of 75% is typically required in SNA for data to be considered reliable [21]. In order to supplement the data collection, the three survey questions were asked again in the semi-structured interviews to collect data not acquired from the surveys. The collected data was then processed using the Gephi 0.9.7 software (available at http://gephi.org) which is an open-source package for social network analysis and visualisation [22]. Network diagrams were created using the Force-Atlas-2 Algorithm, which is a spatialisation algorithm for network visualisation proposed by Gephi team [23]. These diagrams contained nodes representing individuals and ties representing the knowledge flows between them, see for example Fig 3.

## 2.3 Semi-structured interviews

Semi-structured interviews were used to explore subjective viewpoints and to gather in-depth descriptions of the experiences of individuals [24]. A list of interview questions was developed and used in all of the interviews (see S1 Appendix). During the interviews, additional questions were asked based on the direction taken in the interview or to clarify issues. Before starting, the interviewees were reassured about the confidentiality of the process and given background information about the aim of the interview and a brief explanation of the topic. The interviews lasted approximately 50 minutes on average. Face-to-face interviews were audio recorded, and the virtual interviews were conducted on Microsoft Teams and were video recorded. All the interviews were transcribed and thematic analysis performed using the NVivo Mac 1.6.2 software developed by QSR International (available at https://lumivero.com/products/nvivo/). To preserve the anonymity of the interviewees, each interviewee was assigned a number, this number was also used within the social network analysis to represent each node. A thematic analysis approach was subsequently used to analyse the qualitative data collected.

## 2.4 Thematic analysis

A thematic analysis was conducted to analyse the qualitative data collected during the interviews. Thematic analysis is a method for identifying, analysing and reporting patterns within data [25]. The data analysis process followed the work by Saldaña (2009) and Corbin and Strauss (1998) and started with open coding in which the qualitative data was broken down

into discrete parts that were labelled as 'codes' [26, 27]. Codes might be a single word, sentence, paragraph or visualised portion of qualitative data [26]. Then an axial coding was employed to identify connections between codes and assembly them into categories (the axes around which the codes orbit). Finally, the categories were linked together into themes via a reflective process which identified a problem, an issue, or an event that appeared to be significant to the interviewees. Themes exist at a more implicit and abstract level that requires a researcher's interpretation, while categories have explicit content in the form of text and descriptions of the interviewees' statements. Butler-Kisber described the qualitative inquiry process as consisting of extracting verbatim significant statements from the data, formulating meanings about them through the researcher's interpretations, clustering these formulated meanings into a series of organised themes, and elaborating each theme through rich written descriptions [28].

Due to the ethnographic nature of the data, the research methodology was designed to ensure the privacy of participants was protected. A week before each interview, the interviewees were asked to fill in a consent form to confirm their willingness to participate and share information, and to explain the privacy protocols for the data collection, data analysis and storage, and the outputs of the network analysis. All the information contained in the consent form was reconfirmed with the interviewees at the beginning of every interview.

## 3. Results and discussion

### 3.1 Social network analysis results

Three knowledge networks were mapped for each organisation based on the survey and interview responses from the interviewees. These networks were (i) a 'knowledge consultation network' ("whom would you consult when you encounter a knowledge gap?"), (ii) a 'knowledge provision network' ("who would seek help from you when they encounter a knowledge gap?"), and (iii) a 'knowledge creation network' ("who would you communicate with when creating new knowledge?").

These networks for Organisations A and B are shown in Figs 1–3 and some initial observations can be made from these diagrams. The networks for Organisation A appear to have two or three 'main players' in the centre with four or five clusters around a single individual. While Organisation B appears to have networks with four or five 'main players' in the centre with only a couple of clusters around single individuals. These differences might arise because Organisation A has some technology-based knowledge management that enables a more dispersed network whereas Organisation B lacks significant knowledge management technology such that the main players are the principal source of knowledge. It is also noteworthy that Organisation A has a small group of people external to the organisation interspersed in its networks whereas Organisation B has large clusters of external people.

In order to facilitate more detailed interpretation of these networks, five characteristics were then determined to quantitively analyse them, specifically, out-degree centrality, in-degree centrality, betweenness centrality, closeness centrality and network density. Whilst these quantitative measures facilitate comparisons and identification of trends, it should be noted that their values will contain uncertainties arising from interviewees failing to mention some communication pathways for a variety of reasons, including forgetfulness, and the lack of responses from some individuals. However, these uncertainties are unlikely to influence the comparisons and trends discussed below.

Out-degree centrality measures the sum of outbound ties from one node to all adjacent nodes. The direction of the tie is the same as the direction of the knowledge flow, which means the tie starts from the knowledge resource node and points to the knowledge recipient node.

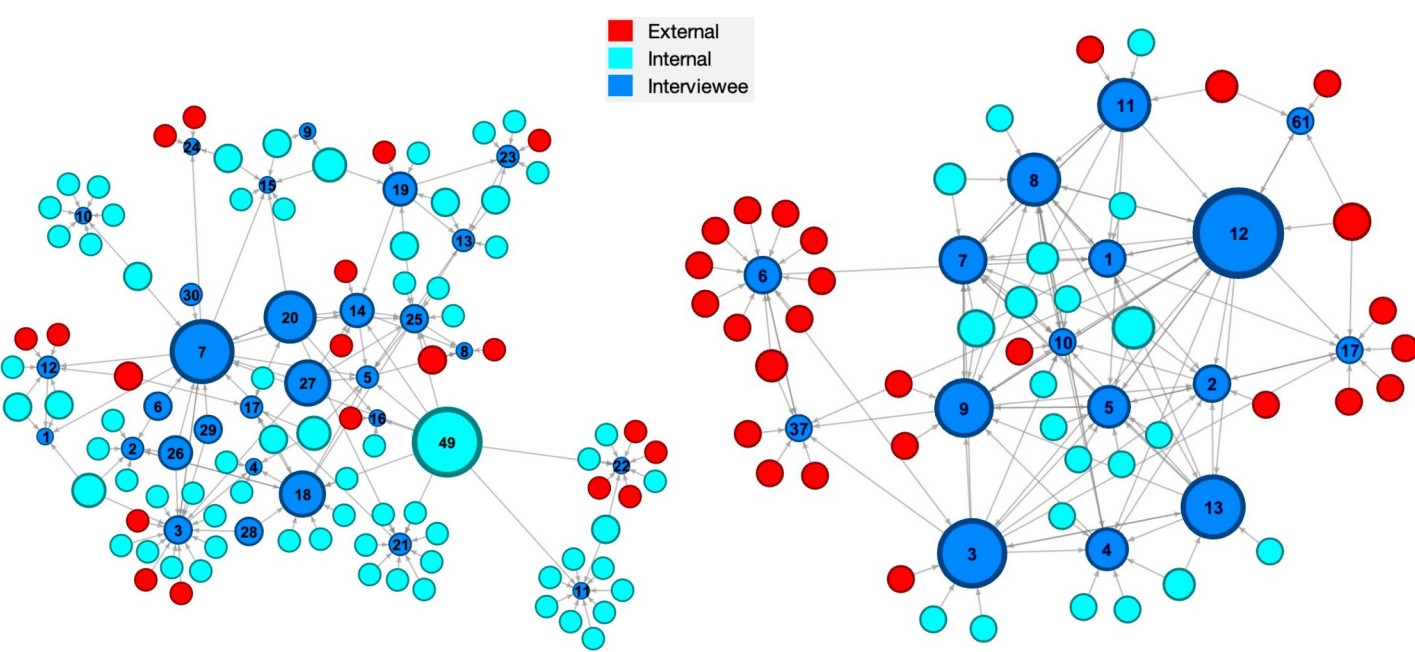

**Fig 1. Knowledge consultation networks.** [Whom would you consult when you encounter a knowledge gap?] for Organisation A [an engineering consultancy] (left) and Organisation B [an electricity generator] (right); the diameter of the nodes indicates the out-degree centrality (sum of outbound ties from a node).

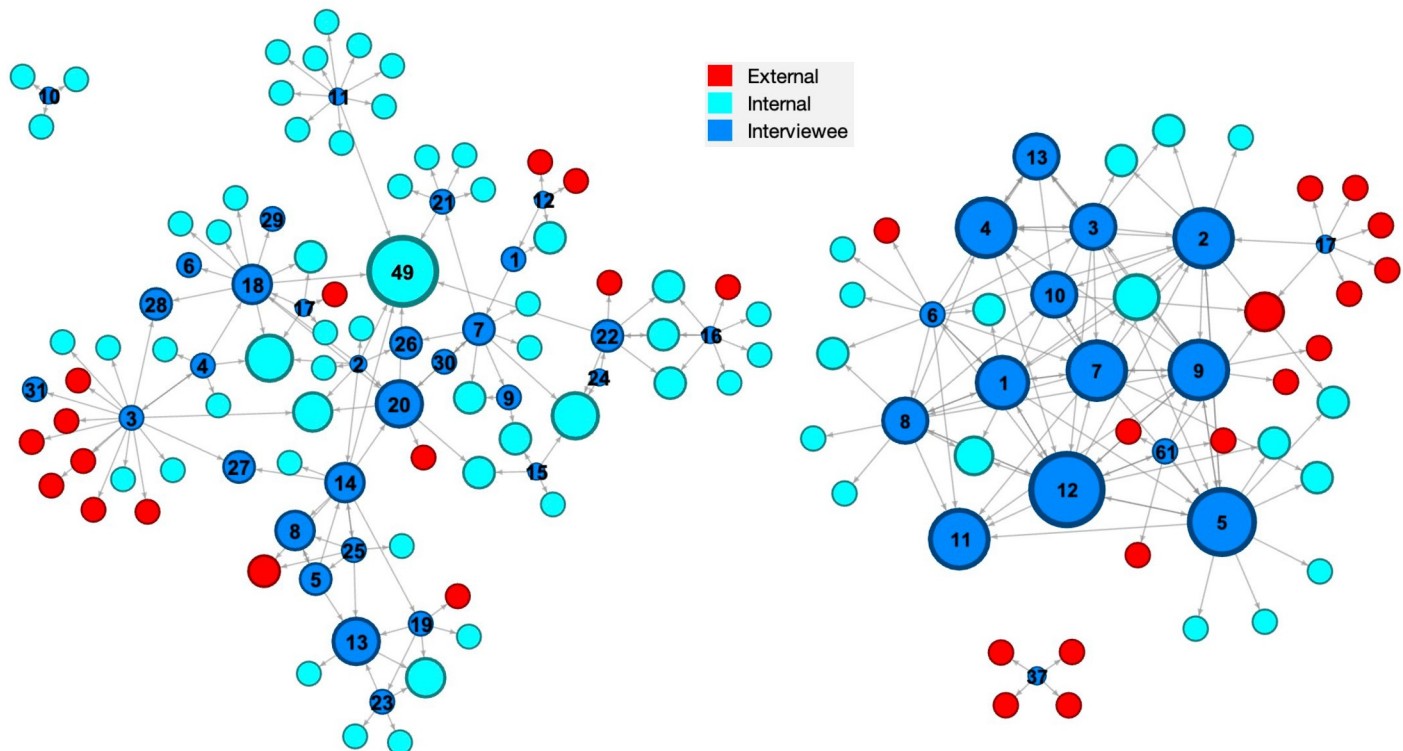

**Fig 2. Knowledge provision networks.** [Who would seek help from you when they encounter a knowledge gap?] in Organisation A [an engineering consultancy] (left) and Organisation B [an electricity generator] (right); the diameter of the nodes indicates the in-degree centrality (sum of inbound ties to a node).

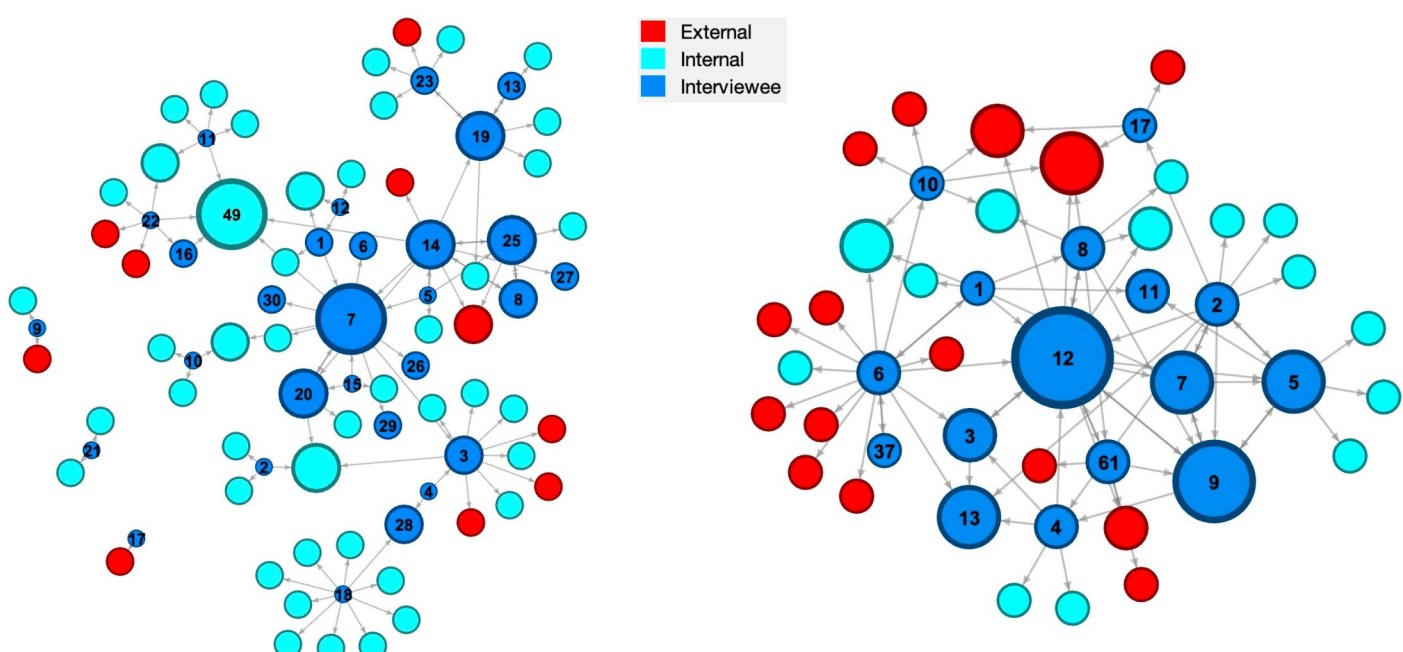

**Fig 3. Knowledge creation networks.** [Who would you communicate with when creating new knowledge?] in Organisation A [an engineering consultancy] (left) and Organisation B [an electricity generator] (right); the diameter of the nodes indicates the in-degree centrality (sum of inbound ties to a node).

In the knowledge consultation network, out-degree centrality is calculated based on how many knowledge flows originate from a node. If an individual, represented by a node, has a high out-degree centrality it means he or she acts as a knowledge resource within the network. In Organisation A, the maximum out-degree centrality is 9 for node 49, which means nine people have identified node 49 as their knowledge resource, while in Organisation B the corresponding result is 13 for node 12. The diameter of a node in Fig 1 indicates its out-degree centrality and Table 1 lists the seven largest values for each organisation in its knowledge consultation network, i.e., the most important sources of knowledge.

In-degree centrality measures the sum of the inbound ties to a node from all adjacent nodes. In the knowledge provision network, in-degree centrality is calculated based on how many knowledge flows are pointing towards a node. The person with a high in-degree centrality acts as a knowledge recipient in the network. In Organisation A the maximum in-degree centrality is 7 for node 49, which means seven people have identified node 49 as their knowledge recipient; while the corresponding result in Organisation B is 8 for node 12. The diameter of a node in Fig 2 indicates its in-degree centrality and Table 2 lists the seven largest values for each organisation in its knowledge provision network, i.e. the largest recipients of knowledge.

The rationale for using out-degree centrality within the knowledge consultation network and in-degree centrality in the knowledge provision network is network data were only collected from interviewees because of the ego-centric approach adopted. Hence the information is uni-directional because in the knowledge consultation network, the data reveals who the interviewees consult when they have a knowledge gap but not who the other nodes consult when they have a similar gap. Similarly, for the knowledge provision network, the interviews reveal who sought knowledge from the interviewees but not who sought knowledge from those who were not interviewed. In other words, in the knowledge consultation network, the knowledge resource was identified by the interviewees and the interviewees considered themselves as knowledge recipients. While in the knowledge provision network, the knowledge

**Table 1. Out-degree centrality (sum of outbound ties for a node) in the knowledge consultation networks in Fig 1 for the individuals with the highest out-degree centrality, i.e., the most important knowledge sources.**

| Organisation A | | Organisation B | |
|---|---|---|---|
| Node No. | Out-degree centrality | Node No. | Out-degree centrality |
| 49 | 9 | 12 | 13 |
| 7 | 8 | 3 | 9 |
| 20 | 6 | 13 | 8 |
| 18 | 5 | 9 | 7 |
| 27 | 5 | 8 | 6 |
| 14 | 3 | 11 | 6 |
| 19 | 3 | 7 | 5 |
| Mean | 5.6 | Mean | 7.7 |

recipient was identified by the interviewees and the interviewees considered themselves as a knowledge resource.

In the knowledge creation network, the node diameter is also based on in-degree centrality. In contrast to the previous two networks, the tie direction is not determined by the knowledge flow. Knowledge creation conversations tend to be two-way communications instead of uni-directional consultations or provisions of knowledge. Hence, the interviewees were asked to identify people who were involved in knowledge creation with them and this determined the tie direction; therefore, the tie directions are from interviewees pointing towards others. Again, the node diameter was proportional to the in-degree centrality in the knowledge creation network in Fig 3, as for the knowledge provision network; hence, the larger nodes have been mentioned more frequently as participants in knowledge creation conversations. Table 3 lists the seven largest values of in-degree centrality for each organisation in knowledge creation network, i.e., the most frequent participants in knowledge creation with the interviewees.

The responses to all three questions described at the start of this section were combined to provide a complete list of people with whom each interviewee interacted and this dataset was used to create the knowledge exchange networks shown in Fig 4. Since, the data are a combination of consultation, provision and creation, the ties have no direction and simply represent an interaction. The diameter of the nodes in this network is proportional to the betweenness centrality which is a measure of the frequency with which a node lies on the shortest path between all other pairs of nodes [29].

A path is an ordered sequence of nodes and ties linking a source node to a target node and its length is the number of ties along it. A node with high betweenness centrality has more

**Table 2. In-degree centrality (sum of inbound ties for a node) in the knowledge provision networks in Fig 2 for individuals with the highest in-degree centrality, i.e., the largest recipients of knowledge.**

| Organisation A | | Organisation B | |
|---|---|---|---|
| Node No. | In-degree centrality | Node No. | In-degree centrality |
| 49 | 7 | 12 | 8 |
| 13 | 4 | 5 | 7 |
| 20 | 4 | 2 | 6 |
| 48 | 4 | 4 | 6 |
| 50 | 4 | 7 | 6 |
| 8 | 3 | 9 | 6 |
| 14 | 3 | 11 | 6 |
| Mean | 4.1 | Mean | 6.4 |

**Table 3. In-degree centrality (sum of inbound ties for a node) in the knowledge creation networks in Fig 3 for the individuals with the highest in-degree centrality, i.e., most frequent participants in knowledge creation conversations with interviewees.**

| Organisation A | | Organisation B | |
| --- | --- | --- | --- |
| Node No. | In-degree centrality | Node No. | In-degree centrality |
| 49 | 5 | 12 | 8 |
| 7 | 5 | 9 | 6 |
| 20 | 3 | 5 | 4 |
| 19 | 3 | 7 | 4 |
| 25 | 3 | 13 | 4 |
| 14 | 3 | 60 | 4 |
| 35 | 3 | 3 | 3 |
| Mean | 3.6 | Mean | 4.7 |

capacity to intervene, either positively or negatively, in interactions between other nodes and, in the knowledge exchange network, to act as a knowledge broker to manage and mediate knowledge flows. While closeness centrality represents the average farness or inverse distance of a node to all the other nodes [30], so that a large value of closeness centrality indicates an ability to reach other nodes via short path lengths and hence quickly disseminate knowledge. Tables 4 and 5 list the nodes with largest values of betweenness centrality and closeness centrality, respectively, for each organisation, i.e., the individuals most capable of acting as interveners and disseminators of knowledge exchange. It is interesting to note that for each organisation, five out of the seven highest value individuals appear in both tables, suggesting that they intervene positively to disseminate knowledge.

The speed or ease with which knowledge travels through a network can be assessed using the average path length, which is 3.77 and 2.75 for Organisation A and B respectively; and, using the network diameter, which is the maximum of the shortest path length between any pair of nodes [30] and is equal to 7 and 4, respectively for Organisation A and B. These values indicate that the network of Organisation A is larger than that of Organisation B in which knowledge transfer likely takes fewer "detours" between two disconnected nodes and therefore disseminates more quickly than in Organisation A. One possible explanation for this finding is the different patterns of work in the two organisations. The interviewees in Organisation A tended to work across different projects which is likely to lead to more widely distributed connections; while those in Organisation B belonged to a single technology group responsibly for a safety critical sub-system who were supported by external suppliers, which would tend to create a more dense and concentrated set of connections.

Network density is the portion of the potential connections in a network that are actual connections and this data is expressed as percentages in Table 6 for the knowledge consultation, provision and creation networks. The network density in Organisation A, the engineering consultancy, is an order of magnitude less than in Organisation B, the electricity generator. In both organisations, the density of the knowledge consultation network is highest and the knowledge creation network has the lowest network density. This suggests that in general, knowledge transferring conversations happen more frequently than knowledge creation conversations.

## 3.2 Semi-structured interview results

Using the social network analysis methodology outlined above, the invisible knowledge communication networks within the two organisations have been revealed. The pre-set questions

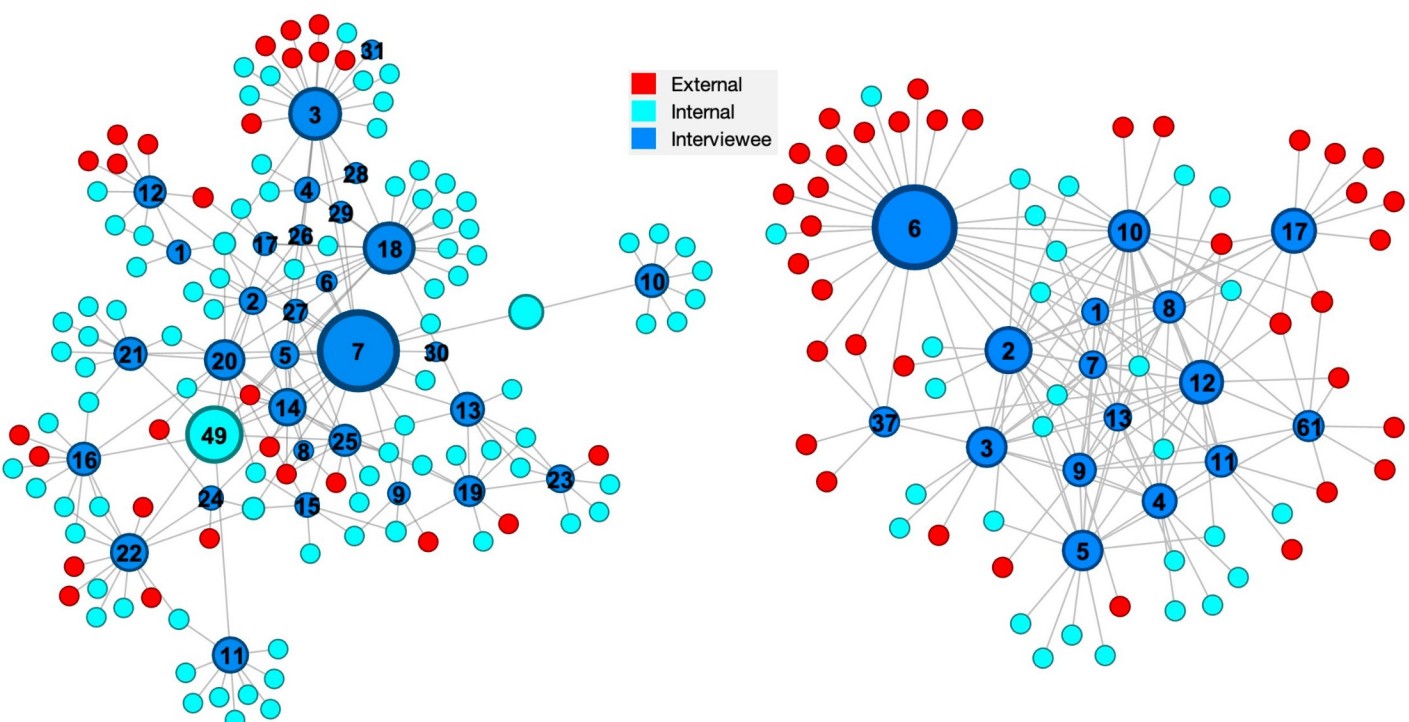

**Fig 4. Knowledge exchange networks.** [A combination of the three respective networks in Figs 1–3] in Organisation A [an engineering consultancy] (left) and Organisation B [an electricity generator] (right); the diameter of the nodes represents the betweenness centrality.

(listed in the S1 Appendix) were used in the interviews to identify the knowledge flowing between people, to explore the broader effectiveness of organisational practices and the adoption of digital technology for knowledge management. A thematic analysis of the transcripts from the interviews was performed from which three themes emerged: (i) forms of knowledge communication; (ii) current knowledge management challenges; and (iii) the potential of digital technology in knowledge management. Each theme is described in the following sub-sections, supported by direct quotes from the transcripts with the organisation and node number given in parentheses after each quote.

**3.2.1 Forms of knowledge communication.** Four forms of knowledge communication were identified from the interview transcripts, specifically: executive decisions, technical

**Table 4. Betweenness centrality (frequency with which a node lies on the short path between other pairs of nodes) in the knowledge exchange networks in Fig 4 for individuals with the highest betweenness centrality, i.e., most likely to act as interveners in knowledge exchange.**

| Organisation A | | Organisation B | |
|---|---|---|---|
| Node No. | Betweenness centrality | Node No. | Betweenness centrality |
| 7 | 4428 | 6 | 1253 |
| 49 | 2676 | 2 | 520 |
| 3 | 2369 | 17 | 481 |
| 18 | 2303 | 12 | 461 |
| 20 | 1488 | 10 | 420 |
| 22 | 1298 | 5 | 394 |
| 14 | 1268 | 3 | 393 |

**Table 5. Closeness centrality (average farness from all other nodes) in the knowledge exchange networks in Fig 4 for individuals with the largest values of close centrality, i.e., those able to disseminate knowledge most quickly.**

| Organisation A | | Organisation B | |
|---|---|---|---|
| Node No. | Closeness centrality | Node No. | Closeness centrality |
| 7 | 0.43 | 12 | 0.57 |
| 49 | 0.42 | 2 | 0.55 |
| 20 | 0.41 | 6 | 0.54 |
| 5 | 0.39 | 3 | 0.53 |
| 14 | 0.39 | 7 | 0.52 |
| 18 | 0.38 | 10 | 0.52 |
| 2 | 0.36 | 8 | 0.50 |

communication, metaknowledge, and information related to task delivery. Executive decisions are requests to a line manager or team leader for their advice, approval, help or support and are driven by the position of the knowledge resource within the formal hierarchy. Technical communication refers to the transmission of expertise possessed by individuals about a relevant technical subject including their experience, judgements and interpretation or intuition as well as their understanding of certain technical issues. Metaknowledge is the 'know-how' that enables people to form connections and communicate with one another and approach the right person for the right information, including knowing who to ask for the knowledge you need and knowing who knows the right people to ask. Information related to task delivery includes, for example, schedules, processes, commercial factors, working relationships with external organisations, and multidisciplinary knowledge across fields associated with the task. Fig 5 shows the distribution of forms of knowledge communication in each organisation based on the thematic analysis.

**3.2.2 Current knowledge management challenges.** There were five categories that were linked together within the theme of current knowledge management challenges. Namely, (a) knowledge management processes are not well-adopted, (b) digital platforms are not well-adopted, (c) digital platforms are not user-friendly, (d) lack of knowledge curation and (e) accessibility of needed knowledge. Each of these categories are discussed below in turn:

a. *Knowledge management processes are not well-adopted*. The interviewees highlighted that existing knowledge management processes were not visible to people, or were not being used. This is likely to lead to knowledge being lost and to the absence of knowledge sharing within the wider organisation. In these circumstances, individuals rely on their personal networks when they are seeking knowledge. Evidence for the existence of this type of situation included statements such as (where the letter and number in parentheses refer to the organisation and interviewee respectively):

○ "It's [a] word of mouth, we speak to people, there's nothing formal." [A14]

○ "We mentioned the pre-job briefing and post-job debriefing. They just ended in the task file. People are not able to find the information if they don't know the task file. Even if you

**Table 6. Network densities (ratio of actual to potential connections expressed as percentages).**

| | Organisation A | Organisation B |
|---|---|---|
| Knowledge consultation network | 0.8 | 2.0 |
| Knowledge provision network | 0.6 | 1.7 |
| Knowledge creation network | 0.4 | 1.3 |

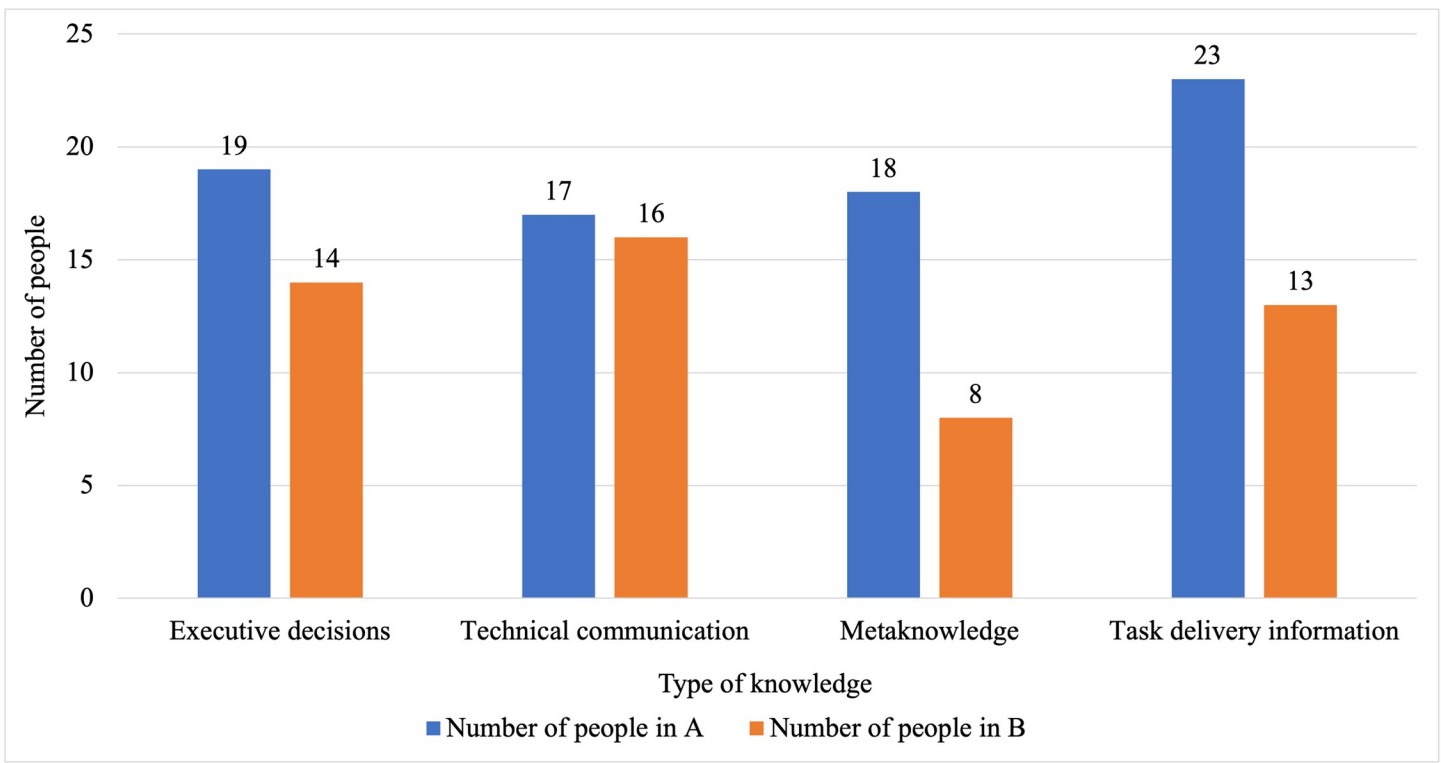

**Fig 5. Distribution of forms of knowledge communication.** Expressed as the number of interviewees who mentioned each type.

knew the task file, you might have struggled to find it because they're not necessarily obvious. They're stored in various different locations on different drives which have different levels of access." [B10]

- "I think we need to learn from experience, but unfortunately, it's not embedded in our natural way of working." [A21]

b. *Digital platforms are not well-adopted.* It was stated frequently in the interviews that digital platforms had been set up within the organisations either with the purpose or potential to enhance knowledge management; however, they were not used regularly by people which would be expected will eventually lead to the systems being underused and redundant. Various reasons were cited for this, including lack of awareness and ignorance of how to use the systems. Examples of statements include:

- "There's a lot of platforms are doing similar things, so I think this is where people will start to get confused between SharePoint, Yammer, and Teams." [A7]

- "But I'm sure there's fountains of knowledge on the enterprise systems that I just actually don't utilise [them]. And I think, again, I'm not necessarily aware of how to utilise it to its full potential." [A24]

c. *Digital platforms are not user-friendly.* The digital platforms that are being used by the organisation were described as user unfriendly due to their outdated functions. This is the most frequently mentioned category in Organisation B. Some examples are:

○ "You do a search, and the document was not there. You go about five minutes later, and you put the same information in, and then it's there. It's impossible." [B5]

○ "The functionality of the systems is probably to a large extent, fine. It's slow and clunky, and I'm sure it can be improved." [B8]

d. *Relying heavily on personal networks.* Apparently due to the lack of well-adopted knowledge management processes and tools, people stated that they rely on their personal networks to find the knowledge and information they need. This causes certain people to become overwhelmed by calls and emails asking for information which could be found on digital systems. This was particularly prevalent when a new recruit or someone in a new role needed to seek or create knowledge. This is one of the most frequently mentioned categories in Organisation A. Some examples include:

○ "It's more about your informal networks, which is fine if you've been here awhile and you've got networks, but actually if you're new, it's more of a challenge." [A29]

○ "We run on networks, people's networks. If somebody leaves, that network disappears, and all that knowledge could be gone." [A26]

e. *Accessibility of needed knowledge.* This involves obtaining tacit knowledge through asking the right questions to the right people and explicit knowledge from digital or paper-based systems easily and efficiently. Interviewees highlighted challenges related to the accessibility of knowledge including issues around searchability and curation of knowledge which were associated with knowledge capture, storage and re-use. Typical comments included:

○ "The system is great, there's so much on it. If you type in the search function, what you want, doesn't always come up. Even though I know it's on there." [A26]

○ "You can fill in a Microsoft Word form to type in your lessons learned, and then you upload the file, but it's not easily accessible, it's not searchable." [A3]

○ "We've got a lot of the historical documents now, but we don't have everything, so occasionally, you will find that you've got most of the drawings, but not all of the drawings you'd like to have. We can't approach people for that information [be]cause they've gone, so we need to go to their organisations, hope they have stored the information." [B3]

**3.2.3 Potential of digital technology in knowledge management.** The interview transcripts provided data related to two research questions associated with digital technology: "what are the digital technologies you have used for knowledge management purposes?" and "what are the expected improvements in terms of digital support for knowledge management?" 'Availability of an intelligent search engine' emerged as a single common category from the thematic analyses of the transcripts from the two organisations and an additional three categories for Organisation A ('using BIM models as knowledge management systems', 'capturing decision making', and 'accessible lessons learned') and two for Organisation B ('availability of an information management system' and 'the need for a communication platform'). These categories are described in the following sub-sections:

a. *Availability of an intelligent search engine.* Interviewees in both organisations mentioned the desirability of having an intelligent search engine for accessing the useful information from the digital systems effectively and for it to be intelligent enough to suggest the appropriate information. For example:

- "If you look at the algorithms that Google run, it's quite clever how they are able to immediately give you answers. I think the key thing with learning from experience is that it needs to exist in a database, and it needs to be keyword driven to be searched. You've got to be able to 'Google' it." [A21]

- "The search engine must be a good place to start. A way forward might be that in some way we digitise all the reports. And then you provide a search engine, like 'Google', to mine that information for anybody." [B3]

The following categories emerged from the transcripts of interviews in Organisation A:

a. *Using BIM models as knowledge management systems*. Building information modelling (BIM) retains knowledge in a digital format about the construction of a building, facilitating updating and sharing of knowledge in the 3D CAD environment [31]. Interviewees mentioned that BIM models have the potential to facilitate shared understanding and support the accessibility of needed information. For example:

- "Some jobs I've used BIM for knowledge management in terms of operation and maintenance of a new plan. You can click on something and it can tell you what the inspection periods are for that piece of plan. Or it can tell you when you need to replace or inspect something." [A14]

b. *Capturing decision making*. The knowledge which supports decision-making is embedded in people's judgments, understanding, conversations, emails, reports and documents. The decision being made can be recorded in documents, for example, in reports or drawings; however, the knowledge behind the decision-making is usually lost in the process. Interviewees mentioned that preserving the knowledge generated during the decision-making process was vital. For example:

- "If they've gone, it's quite difficult to get in touch with people. But what you should try to do in the project is recording as much as you can as you're going along. Every time you make a key decision, you record that decision, and you record why you've made that decision." [A11]

c. *Accessible lessons learned*. The importance of lessons learned and learning from experience was mentioned by many interviewees. Yet in the current work, virtually no lessons learned could be accessed for reuse from the digital systems. Interviewees mentioned their expectations of having lessons learned centrally stored and being searchable and reusable with digital support. For example:

- "If you are at the start of the project, if you put in the keywords, it comes back with learning from experience from other projects. And in my head, the lessons learned should be archived and put into a central depository." [A6]

Two further categories arose from the thematic analysis of the transcripts from in Organisation B: an information management system; and, a communication platform. These are described below:

a. *Availability of an information management system*. A number of interviewees mentioned that an information management system, providing executive information and "know who" knowledge internally and across the supply chain, would be effective in supporting efficient communication and collaboration. An example of their statements is:

 ○ "The use of Wikipedia type pages, it can be helpful to get information through that. Organisations that who does what, and some backgrounds. As a custodian of all the knowledge." [B4]

b. *The need for a communication platform.* A communication platform could work as a common space for knowledge sharing and to facilitate discussions and knowledge creation–this functionality was seen has a useful facility by a number of interviewees. For example:

 ○ "When we started off with Covid, I said what we needed was some sort of equivalent of the coffee room, a chat room that it could, one, take your mind off work for a few minutes, giving you a mental break. Two, it gave you a chance to catch up on what everyone else was doing. Three, the number of times you've got some really interesting technical discussions going on, you go back to your desk afterwards thinking that I've got an idea. There was that cross-fertilisation of ideas." [B12]

### 3.3 Status of knowledge management in organisations A and B

There is an underlying assumption that the adoption of digital technologies by a knowledge intensive organisation will assist developing its knowledge management culture by supporting more rapid and comprehensive collation, curation and creation of knowledge. This normative view is explored in this section using the data acquired and a model that originates in the health and safety ethos of organisations. The evolutionary model of safety culture, described by Hudson in 2001, has been widely used to characterise the development of the ethos of an organisation [17]. The model has five developmental stages from '**pathological**' (who cares as long as we are not caught) through '**reactive**' (safety is important, we do a lot every time we have an accident), '**calculative**' (we have systems in place to manage all hazards) via '**proactive**' (we work on the problems that we still find) to '**generative**' (safety is how we do business round here). When an organisation progresses through these stages people becoming more informed about safety matters and there is an increase in trust associated with the processes.

Two of the co-authors have held responsibility for health and safety in engineering organisations and hence are familiar with both this model and the reality of safety culture. This led to identifying parallels with the evolution of knowledge management which became apparent during this study and hence an evolutionary model of knowledge management has been developed and is shown in Fig 6. As in the evolutionary model of safety culture, there is an increasing awareness as an organisation evolves its knowledge management culture and this is accompanied by increasing integration of knowledge management in people's work processes as well as additive deployment of technology to support knowledge management. These trends are illustrated by the long arrows pointing diagonally upwards and forming a pair of tramlines in Fig 6. Exemplars of the format of the current technologies involved in the additive deployment are shown in the white rectangular boxes to the left of each stage in the evolution.

There are five stages in the new model, as in the model of safety culture that they emulate closely, and they are '**ignored**' (we have no knowledge management and no plans for knowledge management), '**perfunctory**' (we have information management processes regarded as knowledge management), '**compliant**' (we have knowledge management processes because we believe that we should), '**proactive**' (knowledge management is valued and continuously improved) and '**embedded**' (knowledge management is integrated naturally into the daily workflow). In the same way that the evolutionary model of safety culture can act as both a pathway for an organisation wishing to mature its culture and as a diagnostic tool for

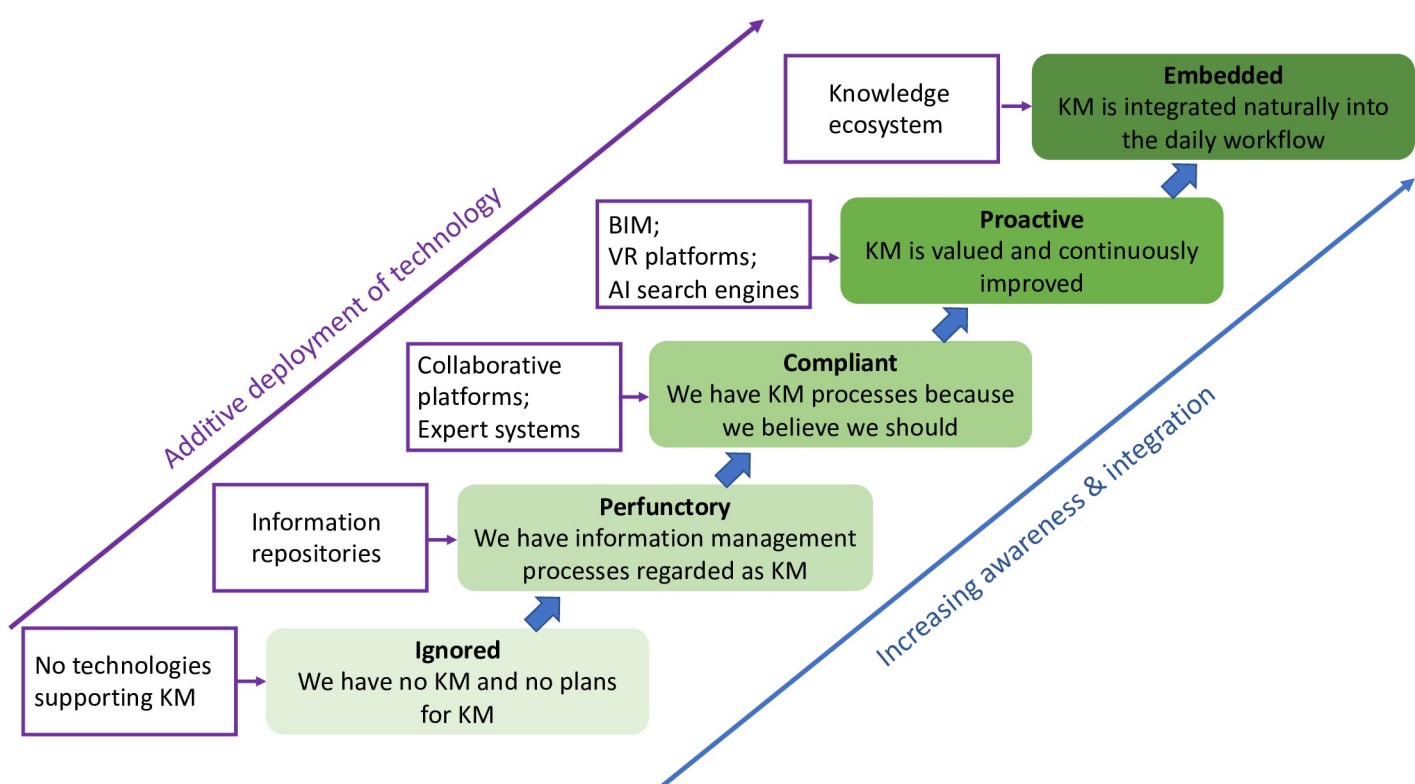

**Fig 6. Evolutionary model of knowledge management (KM).** Based on the evolutionary model of safety culture [17]; the intensity of the colour indicates the level of maturity of the knowledge management culture.

evaluating the status of the safety culture, the new model can act as a pathway and a diagnostic tool for the state of knowledge management in an organisation.

It seems likely the Organisation A in this study is somewhere between 'compliant' and 'proactive', perhaps strategically they are proactive but operationally they are closer to compliant; whereas Organisation B has moved from 'ignored' and might have reached 'perfunctory'. These evaluations are based on the information gained from the semi-structured interviews and therefore are infected with bias; hence, it is interesting to explore whether the level of evolution in knowledge management can be deduced from the social network analysis. The networks for Organisation A tend to have a small number of 'main players' in the centre with a large number of individuals surrounded by a cluster of nodes and relatively low network densities (see Table 4) whereas Organisation B networks have a large number of 'main players' with only a small number with clusters of nodes around them and high network densities–an order of magnitude larger than for Organisation A. This might imply that there are poor or no knowledge management systems in Organisation B with the result that the 'main players' at the centre of the networks are the source of knowledge and it is difficult for individuals to provide and create knowledge independently of them, which is supported by the mean centralities for the highest centrality individuals in the knowledge consultation, provision and creation networks being greater for Organisation B than A (see Tables 1–3). While in Organisation A some level of knowledge management system allows a large number of individuals to access to knowledge and to act independently in providing and creating knowledge both within the organisation and with clients, which is perhaps supported by the larger betweenness centrality

values in organisation A (see Table 4). This supports the conclusion that Organisation A has evolved beyond Organisation B and the ignored or perfunctory stages in Fig 6.

An inevitable question from the management of both organisations is what can we do to improve knowledge management. The evolutionary model would imply that Organisation A should plan and develop a knowledge eco-system which would involve integrating their existing systems and embedding them in the ways of working, probably through training to raise awareness. The results of the thematic analysis would point to similar recommendations with the addition of the need to focus on the tacit knowledge associated with decision-making and lessons learned. For organisation B, training and increasing awareness are also important; however, their focus should be on implementing effective knowledge management systems, such as an expert system and collaboration platform.

It is possible to speculate that further evolution of the knowledge management culture from the stage at which Organisation A has reached would lead to a more dispersed network characterised by a large network diameter (7 for Organisation A compared to 4 for Organisation B) and higher average path length (3.77 for Organisation A compared to 2.75 for Organisation B) with many small clusters or communities of knowledge and low network densities (typical less than one in Organisation A and two to three times greater in Organisation B).

It is recognised that the two case studies do not provide sufficient evidence to draw firm conclusions about the relationship between the morphology of an organisation's knowledge networks and the stage of development of its knowledge management culture. Nevertheless, some trends are apparent from this work as described above and are likely relevant to many engineering organisations that share similar characteristics to either Organisation A and, or Organisation B. It has been assumed that the adoption of digital technologies is essential to maturing knowledge management and this assumption is supported by evidence in the literature that digital technologies are being deployed in five main roles as repositories, transactive memory systems, communication spaces, boundary objects, and non-human actors [8]. Integrating these roles to allow them to be performed simultaneously and seamlessly would deliver the infrastructure for a knowledge management culture with a high level of maturity and would likely provide significant competitive advantage and organisational value to any organisation but will be particularly significant in knowledge intensive sectors, such as engineering.

Finally, the comprehensive implementation of digital technologies in an organisation would make it worthwhile considering the digital technologies supporting knowledge management as nodes in the social analysis network and to explore their impact on the shape of the networks.

## 4. Conclusions

Social network analysis and a series of semi-structured interviews analysed using thematic analysis have been used to investigate the culture associated with knowledge management and the deployment of digital technology to support knowledge management in engineering organisations. One organisation was an engineering consultancy and the other an electricity generator. The results demonstrate that the two engineering organisations had different cultures and levels of engagement with knowledge management which were characterised by different social networks describing knowledge consultation, provision and creation. The engineering consultancy had networks with relatively low network densities and a small number of key nodes surrounded by nodes in turn surrounded by clusters of nodes; whereas the electricity generator had networks with large numbers of key nodes and high network densities. An evolutionary model of knowledge management culture has been proposed based on an existing evolutionary model for safety culture and it is proposed that the engineering consultancy has

evolved its knowledge management culture further than the electricity generator as a consequence of its greater deployment of digital technology for knowledge management. It is proposed that social networks could be used as an indicator of the stage of evolution of knowledge management in engineering organisations with low network density and dispersed networks representing higher stages of evolution.

## Supporting information

**S1 Appendix. Semi-structured interview questions.**
(PDF)

## Acknowledgments

The cooperation and support of Organizations A and B and their employees is gratefully acknowledged. For the purpose of open access, the authors have applied a Creative Commons Attribution (CC BY) license to any published version of this manuscript.

## Author Contributions

**Conceptualization:** Eann A. Patterson, Richard J. Taylor, Yuxin Yao.

**Data curation:** Yuxin Yao.

**Formal analysis:** Eann A. Patterson, Richard J. Taylor, Yuxin Yao.

**Funding acquisition:** Eann A. Patterson, Richard J. Taylor.

**Investigation:** Eann A. Patterson, Richard J. Taylor, Yuxin Yao.

**Methodology:** Eann A. Patterson, Richard J. Taylor, Yuxin Yao.

**Supervision:** Eann A. Patterson, Richard J. Taylor.

**Validation:** Eann A. Patterson, Richard J. Taylor, Yuxin Yao.

**Visualization:** Yuxin Yao.

**Writing – original draft:** Eann A. Patterson, Richard J. Taylor, Yuxin Yao.

**Writing – review & editing:** Eann A. Patterson, Richard J. Taylor, Yuxin Yao.

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
