## [Decision Letter · Decision Letter 0]

10 Oct 2023

PONE-D-23-21893The impact of digital technologies on knowledge networks in two engineering organisationsPLOS ONE

Dear Dr. Yao,

Thank you for submitting your manuscript to PLOS ONE. After careful consideration, we feel that it has merit but does not fully meet PLOS ONE’s publication criteria as it currently stands. Therefore, we invite you to submit a revised version of the manuscript that addresses the points raised during the review process.

Please carefully check the Reviewers’ comments and improve the manuscript. Reviews provide details into areas that require improvement.

We look forward to receiving your revised manuscript.

Kind regards,

Agnieszka Konys, Ph.D.

Academic Editor

PLOS ONE

“This research was jointly funded by the EPSRC Centre for Doctoral Training in NGN (Next Generation Nuclear) and the University of Manchester.”

“This research was jointly funded by the EPSRC Centre for Doctoral Training in NGN (Next

Generation Nuclear) and the University of Manchester. The cooperation and support of

Organizations A and B and their employees is gratefully acknowledged.”

“This research was jointly funded by the EPSRC Centre for Doctoral Training in NGN (Next Generation Nuclear) and the University of Manchester.”

Reviewers' comments:

Reviewer's Responses to Questions

**Comments to the Author**

1. Is the manuscript technically sound, and do the data support the conclusions?

Reviewer #1: Partly

Reviewer #2: Yes

2. Has the statistical analysis been performed appropriately and rigorously? 

Reviewer #1: N/A

Reviewer #2: Yes

3. Have the authors made all data underlying the findings in their manuscript fully available?

Reviewer #1: Yes

Reviewer #2: Yes

4. Is the manuscript presented in an intelligible fashion and written in standard English?

Reviewer #1: Yes

Reviewer #2: Yes

5. Review Comments to the Author

Reviewer #1: Abstract and Keywords

1. The Abstract overstates the study. So, there is the phrase “knowledge is the foundation of… competitive advantage and the primary driver of… value”. This is a very large claim, even for knowledge intensive organisations. Some contextualisation and moderation of this claim would help e.g. “The management and exploitation of knowledge can contribute significantly to value…”

2. The statement “provided insights into the maturity of the knowledge management culture” somewhat presupposes some kind of model of maturation. This in turn implies that relating the findings to Hudson’s safety management culture model was rather more fundamental to the development of the study. (Currently the connection to Hudson reads as a post-analytic intuition). As a reader I felt that Hudson’s model should be described sooner, and perhaps mentioned in the Abstract.

3. There is a claim that a new evolutionary model knowledge management model is proposed. But there are only two sites, and in neither of them does there appear to be data on the evolution or change in knowledge management.

4. I think the keywords should include one or both of “knowledge intensive organisations” and “engineering organisations”, as the findings are specific to these kinds of organisation.

General Comments

5. I think you need to be clearer throughout the piece that this study is related to engineering organisations. Knowledge management is widespread across many sectors, but the specific expectations are very different in different sectors. I am thinking of the knowledge management needs of health care, law firms, investments banks, biotechnology companies, to name a few. All of these need to manage knowledge, but the meaning of “knowledge” differs and so their precise knowledge management needs differ greatly.

6. It is not incorrect to state that this is a mixed methods study, but the questions asked are very different in the two parts. In the interviews, you are interested in digital methods of knowledge management; in the social network analysis, you are interested in who talks to whom. I think there needs to be more connection between these two methods.

7. Some of the numbers in the text do not match the numbers in the figures and tables. E.g. table 1, Node 12 has out-degree centrality of 13, but this is 10 in the text. Table 2 has a similar issue – is node 12’s in-degree centrality 7 or 8?

8. On page 10 the topic of “knowledge creation” is introduced. It would be helpful to know precisely what the question was here.

9. I am not sure about the validity of tables 4 and 5. Do you actually know about all the paths in existence? Since some individuals did not respond, it is presumably possible that there are connections between individuals other than those highlighted. This could be clarified.

10. In lines 254 to 258 you highlight the different patterns of work in the two organisations. But this begs many questions, and points to a need for more contextual information to help the reader. For example, some thick description on “engineering consultancy”, “energy company”, “single technology group” would flesh out the contexts. (That is, using more ethnographic data to interpret the social network data. While I appreciate the need to maintain organisational anonymity, even some general level description of each organisation would help). The point being that different work has implications for knowledge management.

11. There is an implied normative view on knowledge management which could be made clearer. So there appears to be an underlying assumption that some kinds of knowledge management are better than others (some are more mature, etc). While this does make some kind of sense, it is treated in a very cursory manner. The connection to the data is not at all clear: for one thing, there is very little in the data about development over time (the interviews were, in effect, cross-sectional). The idea of development might indeed follow from the adaptation of Hudson’s model, but in that case there needs to be more in-depth justification of why a model for “safety culture” can be (should be?) adapted for “knowledge management culture”. In relation to this, the assertion based on the literature review [ref 13] that “adoption of digital technologies is essential to maturing knowledge management” could perhaps form the starting point for the paper.

Reviewer #2: The aim statement of the paper must be clarified to reflect the main outcome of the paper.

It seems that the new evolutionary model of knowledge management is the actual result of the investigation in this paper.

Several key concepts applied in the paper need a brief explanation (as many readers might not be familiar with them). An addition of a Literature Review section would benefit this paper to explain such concepts relevant to this paper, e.g. the concepts of data, information and knowledge, types of knowledge, the knowledge management process/life cycle, difference between information management system and knowledge management system, relevant digital technologies, social networks in organisations, organisational culture and knowledge Management, etc.

Some discussions in the paper about these concepts should be better placed in a Literature Review section.

The title for section two should be Research Methodology (it is more appropriate).

I think it is important to clarify in the paper, section 2.2, that the survey will help to identify the knowledge transfer networks that currently exists in the organisations under study. These networks are currently hidden because they have not been formally mapped before. But the surveys are not used to “construct” the networks (line 86, line 133). The authors may have constructed a model of the knowledge networks, but not the networks themselves. The knowledge networks already exist in the organisations.

In section 3.2.1 you discuss types of knowledge. But you haven’t explained yet the different types of knowledge in knowledge management. In this section, are you talking about knowledge instruments rather than types of knowledge? Are Tacit Knowledge and Explicit Knowledge examples of types of knowledge? You need to clarify these concepts.

In section 3.3, line 462, page 21, you need a better explanation of the traffic light system, and provide an example. How can you measure (qualitative/quantitative assessment) the level of knowledge management activity.

Somewhere at the end of the paper, you need to explicitly link your work and results with achieving competitive advantage (evidence).

Some discussions of your results in section 3.3 could be generalised considering that Organisation A and B may represent typical engineering organisations in specific sectors, and therefore your work is very relevant for organisations with such characteristics. This contribution needs to be strongly highlighted. Separate your discussions on limitations from your discussions on proposals of further study.

6. PLOS authors have the option to publish the peer review history of their article (what does this mean?). If published, this will include your full peer review and any attached files.

Reviewer #1: No

Reviewer #2: **Yes: **Jose Eduardo Munive-Hernandez

---

## [Author Response · Author response to Decision Letter 0]

14 Nov 2023

Reviewer #1:

Abstract and Keywords

1. The Abstract overstates the study. So, there is the phrase “knowledge is the foundation of… competitive advantage and the primary driver of… value”. This is a very large claim, even for knowledge intensive organisations. Some contextualisation and moderation of this claim would help e.g. “The management and exploitation of knowledge can contribute significantly to value…”

We have modified the opening statement in line with the reviewer’s suggestion.

2. The statement “provided insights into the maturity of the knowledge management culture” somewhat presupposes some kind of model of maturation. This in turn implies that relating the findings to Hudson’s safety management culture model was rather more fundamental to the development of the study. (Currently the connection to Hudson reads as a post-analytic intuition). As a reader I felt that Hudson’s model should be described sooner, and perhaps mentioned in the Abstract.

This sentence has been recast and Hudson’s model introduced into the abstract as proposed by the reviewer.

3. There is a claim that a new evolutionary model knowledge management model is proposed. But there are only two sites, and in neither of them does there appear to be data on the evolution or change in knowledge management.

The term ‘evolutionary’ derives from Hudson’s model and we stated that we had located the organisations on their journey rather than identifying change. We have amended the text to associate ‘evolutionary’ with Hudson’s model in the abstract.

4. I think the keywords should include one or both of “knowledge intensive organisations” and “engineering organisations”, as the findings are specific to these kinds of organisation.

We have amended the keywords using one of the keywords suggested.

General Comments

5. I think you need to be clearer throughout the piece that this study is related to engineering organisations. Knowledge management is widespread across many sectors, but the specific expectations are very different in different sectors. I am thinking of the knowledge management needs of health care, law firms, investments banks, biotechnology companies, to name a few. All of these need to manage knowledge, but the meaning of “knowledge” differs and so their precise knowledge management needs differ greatly.

We acknowledge that the knowledge management needs of organisations in other sectors might be different; however, we have explicited stated in our title, abstract and introduction that our focus is engineering organisations. We have also been specific in the discussion and conclusions and referred repeatedly to engineering organisations. We would prefer to remain silent on the other types of organisations mentioned by the reviewer because we have no evidence to support a statement. We have changed the conclusions to re-iterate that they relate to engineering organisations.

6. It is not incorrect to state that this is a mixed methods study, but the questions asked are very different in the two parts. In the interviews, you are interested in digital methods of knowledge management; in the social network analysis, you are interested in who talks to whom. I think there needs to be more connection between these two methods.

Thank you for highlighting our overuse of the term ‘mixed-method’. We agree with the reviewer and have restricted its use to the description of the social network analysis only in section 2.2.

7. Some of the numbers in the text do not match the numbers in the figures and tables. E.g. table 1, Node 12 has out-degree centrality of 13, but this is 10 in the text. Table 2 has a similar issue – is node 12’s in-degree centrality 7 or 8?

Thank you for picking up these typographical errors, we have corrected them in the text.

8. On page 10 the topic of “knowledge creation” is introduced. It would be helpful to know precisely what the question was here.

This section has been rewritten to be more explicit about all three questions including the one related to knowledge creation.

9. I am not sure about the validity of tables 4 and 5. Do you actually know about all the paths in existence? Since some individuals did not respond, it is presumably possible that there are connections between individuals other than those highlighted. This could be clarified.

There are likely to be more paths than have been identified in the networks due to interviewees neglecting to mention some contacts as well as the lack of responses from some individuals. This implies that there are uncertainties associated with all of the data. This has been clarified in a statement appended to third paragraph in section 3.1.

10. In lines 254 to 258 you highlight the different patterns of work in the two organisations. But this begs many questions, and points to a need for more contextual information to help the reader. For example, some thick description on “engineering consultancy”, “energy company”, “single technology group” would flesh out the contexts. (That is, using more ethnographic data to interpret the social network data. While I appreciate the need to maintain organisational anonymity, even some general level description of each organisation would help). The point being that different work has implications for knowledge management.

We agree that some more context would be helpful and included it in a first draft that we shared with the two organisations. Unfortunately, both organisations asked us to revise the descriptions of them to the statements in the submitted manuscript. We are unwilling to re-open this negotiation which could take many months to conclude, possibly with no substantive changes to the descriptions.

11. There is an implied normative view on knowledge management which could be made clearer. So there appears to be an underlying assumption that some kinds of knowledge management are better than others (some are more mature, etc). While this does make some kind of sense, it is treated in a very cursory manner. The connection to the data is not at all clear: for one thing, there is very little in the data about development over time (the interviews were, in effect, cross-sectional). The idea of development might indeed follow from the adaptation of Hudson’s model, but in that case there needs to be more in-depth justification of why a model for “safety culture” can be (should be?) adapted for “knowledge management culture”. In relation to this, the assertion based on the literature review [ref 13] that “adoption of digital technologies is essential to maturing knowledge management” could perhaps form the starting point for the paper.

Thank you for highlighting this shortcoming and suggesting the starting point which we have adopted in an addition to the introduction. We have also discussed it prior to introducing Hudson’s model in section 3.3.

Reviewer #2:

1. The aim statement of the paper must be clarified to reflect the main outcome of the paper. It seems that the new evolutionary model of knowledge management is the actual result of the investigation in this paper.

Thank you for this suggestion. The model arose out of the consideration of the results and was not the aim of the investigation so it would be disingenuous to report it as an aim. Nevertheless, we appreciate that it is appropriate to introduce it earlier in the paper and have added a statement at the end of the introduction, as well as in the abstract as requested by reviewer #1.

2. Several key concepts applied in the paper need a brief explanation (as many readers might not be familiar with them). An addition of a Literature Review section would benefit this paper to explain such concepts relevant to this paper, e.g. the concepts of data, information and knowledge, types of knowledge, the knowledge management process/life cycle, difference between information management system and knowledge management system, relevant digital technologies, social networks in organisations, organisational culture and knowledge Management, etc. Some discussions in the paper about these concepts should be better placed in a Literature Review section.

Thank you for this suggestion. We feel that these insertions would significantly lengthen the paper and distract from its principal narrative. Instead we have cited some references where readers can refresh their understanding of these topics.

3. The title for section two should be Research Methodology (it is more appropriate).

We have made this change, as requested.

4. I think it is important to clarify in the paper, section 2.2, that the survey will help to identify the knowledge transfer networks that currently exists in the organisations under study. These networks are currently hidden because they have not been formally mapped before. But the surveys are not used to “construct” the networks (line 86, line 133). The authors may have constructed a model of the knowledge networks, but not the networks themselves. The knowledge networks already exist in the organisations.

Yes, we agree. Thank you for spotting our imprecise use of language which we have corrected.

5. In section 3.2.1 you discuss types of knowledge. But you haven’t explained yet the different types of knowledge in knowledge management. In this section, are you talking about knowledge instruments rather than types of knowledge? Are Tacit Knowledge and Explicit Knowledge examples of types of knowledge? You need to clarify these concepts.

We have not discussed tacit or explicit knowledge because we did not attempt to distinguish these types of knowledge in discussion with our interviewees. However, we acknowledge that our use the term ‘type of knowledge’ naturally leads to these concepts whereas infact we have identified ‘forms of knowledge communication’ – we prefer this terminology to knowledge instruments. We have amended section 3.2.1 and figure 5 accordingly.

6. In section 3.3, line 462, page 21, you need a better explanation of the traffic light system, and provide an example. How can you measure (qualitative/quantitative assessment) the level of knowledge management activity.

We have deleted the reference to the traffic light system and simplified the figure.

7. Somewhere at the end of the paper, you need to explicitly link your work and results with achieving competitive advantage (evidence).

We have rewritten the last paragraph of the discussion in order to strengthen this aspect of the work.

8. Some discussions of your results in section 3.3 could be generalised considering that Organisation A and B may represent typical engineering organisations in specific sectors, and therefore your work is very relevant for organisations with such characteristics. This contribution needs to be strongly highlighted. Separate your discussions on limitations from your discussions on proposals of further study.

We have strengthened the point about other organizations and also separated the comment about future work, as suggested.

---

## [Editor Report · Decision Letter 1]

20 Nov 2023

The impact of digital technologies on knowledge networks in two engineering organisations

PONE-D-23-21893R1

Dear Dr. Yao,

We’re pleased to inform you that your manuscript has been judged scientifically suitable for publication and will be formally accepted for publication once it meets all outstanding technical requirements.

Kind regards,

Agnieszka Konys, Ph.D.

Academic Editor

PLOS ONE
---

## [Editor Report · Acceptance letter]

11 Dec 2023

PONE-D-23-21893R1 

The Impact of Digital Technologies on Knowledge Networks in Two Engineering Organisations 

Dear Dr. Yao:

I'm pleased to inform you that your manuscript has been deemed suitable for publication in PLOS ONE. Congratulations! Your manuscript is now with our production department. 

Kind regards, 

on behalf of

Dr. Agnieszka Konys 

Academic Editor

PLOS ONE